# Bridging the Gap Between Foundation Models and Heterogeneous Federated Learning

## Abstract

Federated learning (FL) offers privacy-preserving decentralized machine learning, optimizing models at edge clients without sharing private data. Simultaneously, foundation models (FMs) have gained traction in the artificial intelligence (AI) community due to their exceptional performance across various tasks. However, integrating FMs into FL presents challenges, primarily due to their substantial size and intensive resource requirements. This is especially true when considering the resource heterogeneity in edge FL systems. We present an adaptive framework for Resource-aware Federated Foundation Models (RaFFM) to address these challenges. RaFFM introduces specialized model compression algorithms tailored for FL scenarios, such as salient parameter prioritization and high-performance subnetwork extraction. These algorithms enable dynamic scaling of given transformer-based FMs to fit heterogeneous resource constraints at the network edge during both FL's optimization and deployment stages. Experimental results demonstrate that RaFFM shows significant superiority in resource utilization efficiency and uses fewer resources to deploy FMs to FL. Despite the lower resource consumption, target models optimized by RaFFM achieve performance on par with traditional FL methods applied to full-sized FMs. This is evident across tasks in both natural language processing and computer vision domains.

## 1 Introduction

Federated learning (FL) (McMahan et al., 2017) represents a significant advancement in machine learning, emphasizing decentralized training while preserving data privacy. FL enhances data privacy and collaboration compared to traditional machine learning by enabling model training across multitudes of decentralized devices without direct data sharing. However, challenges like non-identical independent distribution (non-IID) data and heterogeneous computational resources among devices present potential training failures.

Concurrently, transformer-based foundation models (FMs) (Bommasani et al., 2021), typified by GPT (Radford et al., 2019; Brown et al., 2020a; OpenAI, 2023), BERT (Devlin et al., 2018), and ViT (Dosovitskiy et al., 2020), pre-trained on large-scale datasets, have revolutionized AI research. FMs leverage their inherent pre-trained knowledge, achieving exceptional performance across multiple domains in downstream tasks even with limited fine-tuning data.

Given the superior strengths of FMs in few-shot transfer learning, they appear well-suited for non-IID FL environments. However, seamlessly integrating FMs into FL presents significant challenges. The substantial size and intensive resource demands of FMs make their deployment on resource-constrained FL edge devices problematic. Furthermore, the uneven distribution of computational resources within FL increases the difficulty of existing challenges. A resource-limited device must first satisfy the resource requirements for FM optimization, despite the presence of more capable devices within the same FL network, leading to high system requirements overall. Additionally, fine-tuning FMs typically requires approximately seven times the resources compared to inference. This disparity means that FL often faces resource-hungry during model training while leaving resources underutilized during inference.

We propose a framework, Resource-aware Federated Foundation Models (RaFFM), to address the resource utilization challenges in FL. RaFFM uses specialized transformer-based FM compression algorithms tailored for FL-edge environments, and dynamically deploys resource-aware scaled FMs

to local clients. Since FMs are over-parameterized, a subset of salient parameters is more impactful to model performance. RaFFM first identifies and prioritizes the salient parameters in the given FM before FL communication starts. The salient parameter prioritization can ease the resource-aware sub-model extractions from FMs and model fusion in FL global updates via parallel matrix operations. Then, RaFFM generates resource-aware sub-models with salient weights so clients can proceed with the FL fine-tuning cycle. Post-training, given that model inference is more resource-friendly than training, RaFFM can deploy larger models, ensuring optimized performance and consistent resource allocation based on the client's capabilities.

In essence, RaFFM brings forth the following key contributions:

- Designed specialized FM compression algorithms tailored for edge FL.

- Proposed specialized salient parameter prioritization strategy for transformer-based FMs.

- Enhanced resource utilization throughout the training and deployment stages of FL.

- Significant reduction in communication overhead between edge devices and central servers.

## 2 BACKGROUND

### 2.1 FEDERATED LEARNING

With growing concerns over data privacy, Federated Learning (FL) has emerged as a decentralized, privacy-preserving machine learning paradigm. It allows model training on private user data without compromising its confidentiality (McMahan et al., 2017). In FL, private data remains on local clients, and the target model is optimized locally, ensuring data privacy and security. Clients only share model updates, such as weights and gradients, asynchronously, minimizing the risk of data leaks. A representative FL algorithm is FedAvg (McMahan et al., 2017). The innate privacy features of FL have made it a preferred choice in various applications, especially in sectors with stringent privacy requirements like healthcare.

However, data and resource heterogeneity in FL often lead to training failures. Unbalanced training across clients leads to poor model convergence and performance. Recent work in FL has focused on improving gradient descent to stabilize training (Liu et al., 2020; Karimireddy et al., 2020; Li et al., 2020); personalizing model weights to enhance performance on downstream tasks (Deng et al., 2020; Tan et al., 2022); and employing model compression techniques like knowledge distillation, dynamic dropout, and adaptive pruning to reduce overfitting on non-IID datasets and improve communication efficiency (Jiang et al., 2022; Yu et al., 2021; Lin et al., 2020; Yu et al., 2022b). Despite these advances, there remains a gap between traditional model training and FL, particularly in heterogeneous FL-edge environments.

### 2.2 FOUNDATION MODELS

Foundation models (FMs) (Bommasani et al., 2021), such as the GPT family (Brown et al., 2020a; Radford et al., 2019), LLaMA Touvron et al. (2023), ViT (Dosovitskiy et al., 2020), CLIP (Radford et al., 2021), and BERT (Devlin et al., 2018), stand at the forefront of AI advancements. FMs pre-trained on vast datasets exhibiting remarkable performance across multiple tasks. The typical life-cycle of an FM encompasses pre-training, fine-tuning, and application. During pre-training, models undergo unsupervised or self-supervised learning on large datasets. The fine-tuning phase tailors them for specific tasks. As an illustration, GPT models (Brown et al., 2020a; Radford et al., 2019; OpenAI, 2023) acquire grammar, syntax, and semantics during pre-training, making subsequent fine-tuning for tasks like text classification or sentiment analysis more effective. FMs excel in few-shot transfer learning (Brown et al., 2020b), making them particularly suited for data-heterogeneous FL environments where limited and imbalanced local data are present. However, the inherent large size and resource-hungry of FMs pose significant challenges to seamlessly apply in FL settings.

Figure 1: Resource-aware Federated Foundation Model (RaFFM) Overview.

## 3 METHODOLOGY

This section offers an in-depth overview of the primary components of the proposed Resource-Aware Federated Foundation Model (RaFFM): foundation model scaling and resource-aware federated learning.

### 3.1 FOUNDATION MODEL SCALING

RaFFM employs a foundation model scaling technique, inspired by (Muñoz et al., 2022), primarily designed to compress pre-trained FMs while ensuring adherence to the heterogeneous resource constraints in edge-FL systems. The overarching objective is to enhance resource utilization both during the training and inference phases in FL. As highlighted in Figure 1, foundation model scaling incorporates two key components: salient parameter prioritization and high-performance sub-model extraction.

#### 3.1.1 SALIENT PARAMETER PRIORITIZATION

Recent advancements in model compression, such as network pruning (He et al., 2018; Blalock et al., 2020; Yu et al., 2022a) and neural architecture search (White et al., 2023; Muñoz et al., 2022), underscore that deep neural networks, particularly pre-trained FMs, often exhibit over-parameterization. Only a subset of parameters critically influence the model's performance. Identifying these impactful parameters is of paramount importance in resource-limited FL settings. Within RaFFM, we leverage salient parameter prioritization to identify salient parameters in FMs, and extract high-performance sub-models with salient parameters that are uniquely tailored for individual clients' resources. Additionally, salient parameter prioritization can ease the resource-aware sub-model extractions and model fusion in FL global updates using parallel matrix slicing.

**Parameter Salience Score.** Inspired by magnitude-based pruning techniques, salient parameters are recognized by ranking model weights using a salience evaluation metric (examples include the L1 and L2 norms (Li et al., 2016; Kumar et al., 2021)). Our experimental analysis preferred the L1 norm, thus adopted for our context. The L1 norm salience for a channel $c$ within weight matrix $W$ is illustrated by:

$$\text{Salience}(c) = \sum_{i=1}^{n} |W_{c,i}| \tag{1}$$

Equation 1 captures the L1 norm by aggregating the absolute weight values in a channel, reflecting the channel's cumulative significance.

**Salient Parameter Prioritization.** We rank the original model's weight channels based on their salience scores. Prioritization ensures that channels with the highest salience are forefronted in the weight matrix.

$$c_{ranked} = \text{argsort}(\text{Salience}(c), \text{descending}) \tag{2}$$

$$W_{ranked} = W[c_{ranked} :] \tag{3}$$

Equation 2 and Equation 3 delineate the procedure for ranking channels based on salience scores.

#### 3.1.2 PARAMETER PRIORITIZATION IN TRANSFORMER

While existing magnitude-based compression methods predominantly target convolutional neural networks (CNNs), applying salient parameter prioritization to multi-head attention transformers ar-

chitectures (Vaswani et al., 2017) requires additional deliberation. We introduce a specialized salient parameter prioritization strategy tailored for transformers, which ensures the preservation of the inherent attention information present in the original FM.

$$\text{Attention}(W^q, W^k, W^v, x) = \text{softmax}\left(\frac{xW^q(xW^k)^T}{\sqrt{d_k}}\right)xW^v \tag{4}$$

Equation 4 characterizes the attention-head. Here, the input sequence $x \in \mathbb{R}^{l \times d}$ has a sequence length of $l$ and an embedding size of $d$. The matrices $W^q \in \mathbb{R}^{d \times d_k}$ and $W^k \in \mathbb{R}^{d \times d_k}$ represent the query and key weights, respectively, while $W^v \in \mathbb{R}^{d \times d_v}$ stands for the value weights.

**Theorem 1**: Given matrices $W^q$ and $W^k$ of dimensions $d \times d_k$ and an input $x$ of size $l \times d$, if we uniformly apply a permutation $\pi$ to the columns of both $W^q$ and $W^k$ to obtain $W^{q'}$ and $W^{k'}$ respectively, the subsequent dot-product attention scores determined using $W^{q'}$ and $W^{k'}$ will match those derived using the original $W^q$ and $W^k$. (A detailed proof is provided in Appendix A.1).

$$\text{Salience}(c) = \frac{\text{Salience}(w_c^q) + \text{Salience}(w_c^k)}{2} \tag{5}$$

To retain the inherent attention characteristics of FMs, we prioritize $W^q$ and $W^k$ within an attention head by employing the same ranked permutation. The salience score, defined in Equation 5, calculates the average norm of the query and key matrices for each channel index $c$. A consistent salience-based rank is concurrently imposed on both $W^q$ and $W^k$. As validated by **Theorem 1**, this ensures that post-prioritization, the derived dot-product attention scores will remain identical with the original attention head of the FMs.

### 3.1.3 HIGH-PERFORMANCE SUB-MODEL EXTRACTION

Given a resource constraint, denoted as $\tau$, and a FM $\mathcal{F}(\mathcal{W})$ with weights $\mathcal{W}$, the objective of high-performance sub-model extraction is to derive a sub-model from $\mathcal{F}(\mathcal{W})$ that comprises salient parameters and adheres to constraints $\tau$.

Leveraging the salient parameter prioritization, we can easily extract high-performance sub-networks using weight matrix slicing. We represent a sub-model, $\mathcal{F}(\mathcal{W}_{c_\tau})$, as a network configuration $c_\tau \in \mathbb{R}^n$ satisfying constraint $\tau$. The $c_\tau$ is a list specifying the layer width for each hidden layer of the FM. To sample the target $c_\tau$, the sampling space is denoted by $\mathcal{S} = [s_1, ..., s_n]$ where $s_i$ denotes the width (number of channels) of the $i^{th}$ hidden layer.

For a sampled network configuration $c_\tau$, we evaluate the size of the sub-model $\mathcal{F}(\mathcal{W}_{c_\tau})$ in terms of measurable metrics (e.g., number of parameters). If it aligns the constraints $\tau$, we extract $\mathcal{F}(\mathcal{W}_{c_\tau})$ using Equation 6:

$$\mathcal{W}_{c_\tau} = \mathcal{W}[: c_\tau] = \{w_i[: c_i] | w_i \in \mathcal{W}, c_i \in c_\tau, \text{and } i = 1, ..., n\} \tag{6}$$

In the above equation, $w_i$ represents the weights of the $i^{th}$ hidden layer, and $c_i$ stands for the number of channels sampled from the $i^{th}$ hidden layer. The salient parameter prioritization component ensures the extracted sub-model, $\mathcal{F}(\mathcal{W}_{c_\tau})$, retains the most salient weights from the original FM.

### 3.2 RESOURCE-AWARE FEDERATED FOUNDATION MODELS

RaFFM can seamlessly integrate with mainstream FL model fusion algorithms, such as FedAvg (McMahan et al., 2017) and Fedprox (Li et al., 2020). This is because all resource-aware local models in RaFFM are sub-networks derived from the forefront channels of the given FM. As a result, heterogeneous federated learning model fusion can be easily executed using matrix slicing, as demonstrated in Equations 6 and 7. In this paper, we use FedAvg as our backend algorithm.

$$\mathcal{W}^t = \sum_{c_\tau \in C} (\mathcal{W}^{t-1}[: c_\tau] + \eta_{c_\tau} \nabla \mathcal{W}_{c_\tau}^t) \tag{7}$$

Table 1: Results on GLUE benchmark.

| Model | Method | #Param | QQP | QNLI | SST-2 | CoLA | STS-B | MRPC | RTE | MNLI | AVG | Training Accel. |
|---|---|---|---|---|---|---|---|---|---|---|---|---|
| BERT-Large | FL | 335M | 89.01 | 91.46 | 93.81 | 56.08 | 89.61 | 75.24 | 68.59 | 85.71 | 81.19 | 1.00 × |
| | RaFFM | **100M** | 88.40 | 91.18 | 94.15 | 54.72 | 88.81 | 75.49 | 68.23 | 85.40 | 80.80 | **6.13 ×** |
| BERT-Base | FL | 109M | 88.34 | 86.43 | 91.74 | 48.03 | 86.34 | 77.94 | 64.26 | 80.83 | 77.99 | 1.00 × |
| | RaFFM | **60M** | 88.00 | 88.41 | 91.63 | 47.76 | 86.00 | 77.45 | 63.90 | 80.33 | 77.94 | **2.45 ×** |
| RoBERTa | FL | 125M | 89.19 | 91.19 | 93.92 | 49.68 | 87.11 | 84.80 | 71.48 | 86.20 | 81.70 | 1.00 × |
| | RaFFM | **53M** | 89.14 | 91.25 | 93.92 | 52.20 | 87.67 | 85.29 | 65.34 | 86.22 | 81.38 | **3.62 ×** |
| DistilBERT | FL | 67M | 86.73 | 84.71 | 89.91 | 45.36 | 83.76 | 79.17 | 55.60 | 77.90 | 75.39 | 1.00 × |
| | RaFFM | **50M** | 86.80 | 85.08 | 90.48 | 41.69 | 82.85 | 78.19 | 57.40 | 78.34 | 75.10 | **1.55 ×** |

Equation 7 aggregate resource-aware local models, $c_\tau$ represent the local model configuration satisfying constraint $\tau$, and $\eta_{c_\tau}$ signifies the learning step for the client.

The RaFFM procedure in an FL communication round can be described as follows:

1. In the $t^{th}$ communication round, RaFFM first perform salient parameters prioritization for the global FM $\mathcal{F}(\mathcal{W}^t)$.

2. RaFFM samples a set of sub-network configurations, $C = \{c_{\tau 1}, .., c_{\tau k}\}$, in accordance with the resource constraints of participating clients, then generate sub-models.

3. These sampled sub-models, represented as $\{\mathcal{W}^t_{c_\tau} | c_\tau \in C\}$, are dispatched to the local participating clients for further training.

4. Once local training finished, the clients relay their model updates back to the server.

5. The server then undertakes model fusion using Equation 7.

The entire process is iteratively executed until federated learning is complete (Refer Appendix 1).

## 4 EXPERIMENTS

### 4.1 EXPERIMENT SETUP

**Federated Learning Settings.** RaFFM is designed to address the challenges posed by resource heterogeneity in Federated Learning (FL) scenarios, especially deploying resource-hungry FMs. Our experiments were conducted in a standard cross-silo FL setup. This environment comprises 100 local clients coordinated by a central server. In each communication round, 10% of the clients are randomly selected to participate in local updates. We employ FedAvg (McMahan et al., 2017) as the underlying FL algorithm for model aggregation.

**Model and Datasets.** Our evaluations span a variety of pre-trained models:

- NLP models: DistilBert (Sanh et al., 2019), RoBERTa (Liu et al., 2019), BERT (Devlin et al., 2018), and FLAN-T5 (Chung et al., 2022).
- Large language model: LLaMA2 (Touvron et al., 2023).
- Computer Vision model: ViT (Dosovitskiy et al., 2020).

For our experiments, we employ diverse datasets, including the GLUE benchmark (Wang et al., 2018), question-answering benchmarks like SQuAD and SQuAD-v2 (Rajpurkar et al., 2016), SVAMP (Patel et al., 2021), CIFAR-10 and CIFAR-100 (Krizhevsky & Hinton, 2009), and the Flower-102 dataset (Nilsback & Zisserman, 2008).

Table 2: Experiments on question answering tasks

| Model | Method | SQuAD | | | | SQuAD-V2 | | | |
|---|---|---|---|---|---|---|---|---|---|
| | | #Param | Exact Match | F1 | Training Accel. | #Param | Exact Match | F1 | Training Accel. |
| BERT-Large | FL | 334M | 83.29 | 89.25 | 1.00 × | 335M | 82.05 | 90.24 | 1.00 × |
| | RaFFM | **95M** | 83.34 | 90.87 | **6.59 ×** | **103M** | 82.27 | 90.28 | **6.59 ×** |
| BERT-Base | FL | 109M | 73.68 | 82.99 | 1.00 × | 109M | 72.85 | 82.69 | 1.00 × |
| | RaFFM | **70M** | 72.37 | 82.00 | **1.94 ×** | **73M** | 71.23 | 81.36 | **1.82 ×** |
| DistillBERT | FL | 67M | 71.70 | 81.30 | 1.00 × | 67M | 69.55 | 79.55 | 1.00 × |
| | RaFFM | **34M** | 70.59 | 80.15 | **2.77 ×** | **34M** | 69.60 | 79.75 | **2.77 ×** |
| RoBERTa | FL | 124M | 82.64 | 89.75 | 1.00 × | 124M | 81.05 | 89.08 | 1.00 × |
| | RaFFM | **90M** | 82.42 | 89.71 | **1.61×** | **89M** | 80.77 | 88.83 | **1.64×** |
| FLAN-T5 Small | FL | 77M | 48.65 | 64.43 | 1.00 × | 77M | 49.80 | 61.49 | 1.00 × |
| | RaFFM | **61M** | 48.83 | 64.60 | **1.42×** | **61M** | 50.00 | 61.74 | **1.42×** |
| FLAN-T5 Base | FL | 248M | 60.70 | 77.75 | 1.00 × | 248M | 67.22 | 80.24 | 1.00 × |
| | RaFFM | **163M** | 60.92 | 78.01 | **1.88×** | **164M** | 67.67 | 80.33 | **1.86×** |

## 4.2 LEARNING EFFICIENCY

Our primary objective with RaFFM is to achieve efficient FL by leveraging fewer computational resources without significantly compromising model performance. To this end, we compare the performance of resource-aware sub-models deployed through RaFFM against the conventional full-size FL model implementations.

### 4.2.1 RESULTS ON GLUE BENCHMARK

We assessed RaFFM's efficacy using the NLP General Language Understanding Evaluation (GLUE) benchmark (Wang et al., 2018), which comprises eight language datasets: QQP, QNLI, SST-2, CoLA, STS-B, MRPC, RTE, and MNLI.

In experiments, baseline FL optimizes full-size FMs at local clients. In contrast, RaFFM deployed resource-aware sub-models tailored to individual clients. Table 1 summarizes the results. Performance metrics were computed for the global model post-training on validation datasets. The column titled #Param represents the average count of local model parameters across all clients for the eight datasets. Training acceleration estimates the relative GPU hours. We also present an average score (AVG) as an aggregate performance indicator.

Notably, RaFFM required fewer computational resources (evidenced by the reduced parameter count) and demonstrated faster training times than the conventional full-model FL. Impressively, despite the reduced model size at the client side, RaFFM-optimized models not only competitive but occasionally surpassed the performance of baseline full-size FL models. This superiority was particularly pronounced in datasets such as SST-2, MRPC, and MNLI.

### 4.2.2 RESULTS ON QUESTION ANSWERING BENCHMARK

Further, we evaluate RaFFM on the Stanford Question Answering Dataset (SQuAD and SQuAD-V2) (Rajpurkar et al., 2016).

Table 2 presents the results across various models. Impressively, RaFFM consistently speeds up the training process on both SQuAD tasks. Taking BERT-Large as an example, RaFFM accelerates FL training (measured in GPU hours) by a factor of 6.59. Remarkably, despite deploying substantially trimmed models at edge clients, RaFFM's efficiency advantage does not come at the cost of model performance. Indeed, in cases like FLAN-T5 base and BERT-Large, RaFFM's performance even eclipses that of full-size FL deployments.

Figure 2: (a) parameter efficiency on SQuAD benchmark. (b) parameter efficiency on SQuAD-V2 benchmark. (c) Average communication footprint per communication round. (d) Communication cost in GLUE benchmark

Table 3: Few-shot learning results on Vision Transformer

| Model | Method | Training Accel. | Communication Traffic/Client | #Param | CIFAR-10 | | CIFAR-100 | | Flower102 | |
|-------|--------|-----------------|------------------------------|--------|----------|------|-----------|------|-----------|------|
| | | | | | Accuracy | F1 | Accuracy | F1 | Accuracy | F1 |
| ViT | FL | 1.00× | 360.87MB | 86M | 97.26% | 97.26% | 91.26% | 91.27% | 94.71% | 94.56% |
| | RaFFM | 1.76× | 247.57MB | 59M | 96.27% | 96.26% | 90.37% | 90.39% | 94.29% | 94.09% |

We further evaluate the parameter efficiency—especially vital in resource-tight FL settings. Figure 2 (a) and (b) visually depict RaFFM's supremacy in parameter efficiency (higher values are preferable). As an illustrative point, in BERT-Large, RaFFM, with an average model size of 95M parameters, posts an Exact Match score of 83.34 and an F1 score of 90.87 on SQuAD, which stands in stark contrast to full-size FM deployment in FL, and the efficiency translates to a speed-up of 3.52 times.

## 4.3 RESULTS ON COMPUTER VISION TASKS

In addition to our evaluations on NLP benchmarks, we extended our assessment to ascertain the efficacy of RaFFM on computer vision (CV) tasks, specifically leveraging the Vision Transformer (ViT) (Dosovitskiy et al., 2020). As shown in Table 3, we fine-tune ViT with 12-shot learning. The results corroborate RaFFM's consistent performance metrics on CV tasks. Specifically, in line with our findings from NLP benchmarks, RaFFM demonstrated a marked superiority in the training acceleration. Additionally, the reduced communication overhead and negligible compromise in model performance further underline its efficiency and robustness.

Conclusively, RaFFM offers an excellent balance between performance and computational efficiency. With fewer parameters and higher speed-up factors, it provides a viable alternative to more computationally intensive models without perceptible degradation in performance.

## 4.4 COMMUNICATION EFFICIENCY

A key challenge in FL is the substantial communication cost, often due to frequent model weights or gradients sharing between edge devices and the central server. Figure 2 (c) illustrates the average communication footprint of a client in each round under various FMs. RaFFM substantially reduces communication costs at the client level. The rationale is straightforward: its resource-aware model deployment leads to more compact models at the edge. Consequently, there is less information to relay between the edge devices and the server, reducing communication burdens.

Additionally, we optimized FMs to meet specific performance benchmarks (set to the average median convergence accuracy). We monitored the network traffic induced by the training process during communication. As highlighted in Table 4, it consistently underlines RaFFM's superiority in minimizing communication costs across all experimental setups. Furthermore, Figure 2 (d) showcases the network traffic needed to achieve the average median convergence performance on the GLUE benchmark. RaFFM also consistently outperforms full-size model deployment, evidencing significantly decreased average communication cost across different FMs.

## 4.5 RESOURCE EFFICIENCY

### 4.5.1 SYSTEM RESOURCE EFFICIENCY

Traditional FL approaches often suffer inefficient system resource utilization. When identical models are dispatched to both high-end and resource-constrained devices, the latter often dictates the

Table 4: Communication cost for achieving target performance on QA benchmark

| Model | Method | Dataset | Target Performance (F1 Score) | Communication Cost Traffic/Client | Total | Δ Cost |
|---|---|---|---|---|---|---|
| BERT-Large | FL RaFFM | SQUADv1 | 85% | 1401MB **398MB** | 109.45GB **46.64GB** | -62.81GB |
| | FL RaFFM | SUQADv2 | 90.09% | 1401MB **432MB** | 73.88 GB **24.47 GB** | -49.41 GB |
| DistillBERT | FL RaFFM | SQUADv1 | 80.15% | 281.14MB **142.67MB** | 29.65 GB **27.31 GB** | -2.34 GB |
| | FL RaFFM | SQUADv2 | 79.55% | 281.14MB **142.67MB** | 50.52 GB **25.64 GB** | -24.88 GB |
| FLAN-T5 Base | FL RaFFM | SQUADv1 | 77.28% | 1038.60MB **687.81MB** | 223.14GB **147.78GB** | -75.36GB |
| | FL RaFFM | SQUADv2 | 79.86% | 1038.60MB **688.40MB** | 547.70GB **322.68GB** | -225.02GB |
| FLAN-T5 Small | FL RaFFM | SQUADv1 | 61.45% | 322.39MB **255.32MB** | 88.16GB **54.88GB** | -33.28GB |
| | FL RaFFM | SQUADv2 | 60.89% | 322.39MB **255.48MB** | 75.56GB **69.86GB** | -5.70GB |

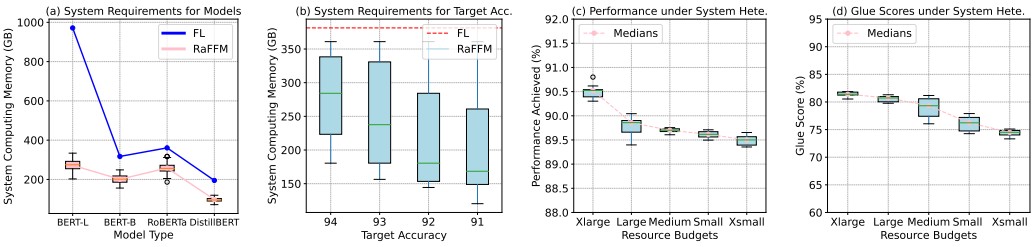

Figure 3: (a) Lowest system resource requirements for deploying various FMs in FL. (b) System requirements for achieving target performance. (c) Question answering performance under various resource budgets. (d) Glue score under various resource budgets.

constraints, compelling the entire system to conform to its limitations. This results in inflated system requirements and often leaves high-end devices underutilized. Figure 3 (a) elucidates this challenge, showcasing the minimum system resource prerequisites for various FMs within a 100-client FL system to achieve the median F1 scores as detailed in Table 2. The baseline FL has lofty uniform resource demands, whereas RaFFM, illustrated through box plots, showcases a range of resource allocations wherein it attains the target performance. Evidently, RaFFM exhibits enhanced system resource efficiency, paving the way for a more cost-effective FL setup in terms of hardware requirements.

Further clarity is provided in Figure 3 (b), which shows the system requirements of RoBERTa deployments for different performance levels on the SST-2 dataset. The red dashed line represents full-size FM deployment in traditional FL, which indicates high baseline system requirements, irrespective of the performance target. Conversely, RaFFM, depicted by the box plot, offers a flexible range of system requirements to attain similar performance outcomes. This adaptability of RaFFM enables dynamic system budget adjustments based on performance goals, maximizing resource efficiency and preventing unnecessary expenditures.

In Figure 3 (c) and (d), we further assess RaFFM's robustness amidst a variety of system resource constraints, spanning Xlarge to XSmall budget categories. The findings underscore RaFFM's consistency: even as resources swell or dwindle, its performance remains steadfast, signifying its adaptability in heterogeneous resource environments.

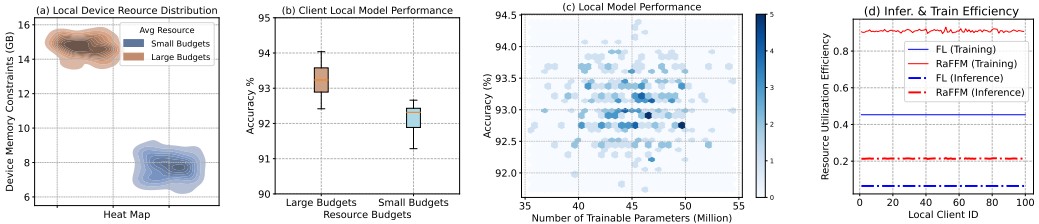

Figure 4: (a) Resource distribution heat map of two distinct FL systems. (b) Client local model performance on SST-2 in distinct FL systems. (c) Heat map of local model performance under different model sizes. (d) Resource utilization efficiency in the training stage and inference stage.

Table 5: Efficient instruction tuning on LLM

| Model | Method | Adapter | Dataset | #Clients | Avg. Memory ↓ | #Trainable Param | Accuracy |
|---|---|---|---|---|---|---|---|
| LLaMA-2 | FL | LoRA | SVAMP | 10 | 1.00 × | 262.41M | 53.44% |
|  | RaFFM |  |  |  | **1.63 ×** | **189.72M** | 52.80% |

### 4.5.2 EDGE RESOURCE EFFICIENCY

In FL, not only do system-wide resource requirements matter, but individual client resources also play a pivotal role. Given the varied resource capacities of individual clients, ensuring stable performance of resource-aware models on these clients becomes crucial.

Figure 4 (a) and (b) depict two distinct resource budget FL systems. The distribution heat map of clients' resources for these two setups is showcased in (a). Figure 4 (b) depicts local clients' performances, demonstrating the stability of local clients' performance: even with notable variations in local resources, every client's performance aligns closely with its peers. This inherent stability is further emphasized when comparing the two different FL systems; despite the system disparities, the local model performance across the two systems remains stable. This consistency is underscored by Figure 4 (c), which plots the relationship between local model size and its achieved performance. Interestingly, even as the model sizes differ, each maintains commendable local performance. To sum up, RaFFM showcases its prowess in adeptly navigating resource heterogeneity, ensuring that performance remains stable at the client level.

### 4.5.3 EDGE INFERENCE EFFICIENCY

The resource consumption during the training phase is usually at least seven times that of the inference phase. This disparity means that FL often faces resource-hungry during model training while leaving resources underutilized during inference.

The mentioned observation is depicted in Figure 4 (d), which delineates the resource utilization efficiency during both training and inference phases for edge clients. RaFFM, with its inherent FM scaling components, enables the post-training deployment of relatively larger models at the edge during the inference stage, hence increasing the model performance and resource utilization.

### 4.6 ENHANCING RaFFM WITH EFFICIENT FINE-TUNING: A CASE WITH LoRA

Incorporating parameter efficient fine-tuning (PEFT) methods like LoRA (Hu et al., 2021) into RaFFM holds potential for optimized performance, especially when dealing with large language models (LLMs) in an FL setting. Though PEFTs such as LoRA can markedly trim the trainable parameters of LLMs—often to less than 1% of the total parameters—it's essential to note that the full-size weights and activations are still retained during training. This results in substantial memory overheads.

To investigate the benefits of this synergy, we paired RaFFM with LoRA and fine-tuned the LLaMA2 model (Touvron et al., 2023) using the preprocessed instruction math question answering dataset, SVAMP (Hu et al., 2023; Patel et al., 2021). This dataset was partitioned among 10 FL clients for the experiments. Table 5 highlights the strengths of the RaFFM-LoRA combination. Specifically, compared to a full-size model paired with LoRA in an FL context, RaFFM coupled with LoRA demonstrates enhanced communication efficiency and a marked acceleration in inference.

## 5 CONCLUSION

We propose RaFFM addressing the challenges when deploy FMs to resource-heterogeneous FL systems. RaFFM introduced specialized FM compression algorithms for edge-FL system that allows scaling down the FM to edge constraints. The experiments demonstrate RaFFM's capability to optimize resource utilization during FL's life cycle, showing its potential for resource-efficient FL. Moreover, the flexibility of RaFFM allows for accelerated LLM fine-tuning in FL with PEFT. Nevertheless, it is essential to recognize the limitations of our approach. Notably, certain foundation models, even post-compression – such as Llama-7B – remain unsuitable for deployment on resource-constrained edge devices in FL settings. Addressing this limitation necessitates advancements in both hardware technology and algorithmic strategies, marking a promising avenue for our future research endeavors.

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

## A  APPENDIX

### A.1  PROOF FOR THEOREM 1

**Notations:** To better illustrate the proof, we decompose $W^q$ and $W^k$ as a set of vectors:

$$W^q = [w_1^q, ..., w_c^q, ..., w_{d_k}^q]$$

and

$$W^k = [w_1^k, ..., w_c^k, ..., w_{d_k}^k]$$

, where $w_c^q, w_c^k \in \mathcal{R}^{d \times 1}$, and $c \in [1, d_k]$. Similarly, $W^{q'}$ and $W^{k'}$ can be decomposed as:

$$W^{q'} = [w_{\pi(1)}^q, ..., w_{\pi(d_k)}^q] \tag{8}$$

$$W^{k'} = [w_{\pi(1)}^k, ..., w_{\pi(d_k)}^k] \tag{9}$$

**Theorem**: Given matrices $W^q$ and $W^k$ of size $d \times d_k$ and an input $x$ of size $n \times d$, if we apply a consistent permutation $\pi$ to the columns of both $W^q$ and $W^k$ to obtain $W^{q'}$ and $W^{k'}$, respectively, then the resulting dot-product attention scores computed using $W^{q'}$ and $W^{k'}$ will be identical to those computed using the original $W^q$ and $W^k$.

**Proof**:

Let $Q = xW^q$ and $K = xW^k$ be the original query and key matrices. Their sizes are $n \times d_k$.

Let $Q' = xW^{q'}$ and $K' = xW^{k'}$ be the query and key matrices after applying the permutation $\pi$.

By our definition of the permutation, for each column $c$ in $W^q$ and $W^k$, the new position in the permuted matrices is $\pi(c)$. So, the elements of $Q'$ and $K'$ are given by:

$$Q'_{ij} = \sum_{c=1}^{d} x_{ic} W^{q'}_{cj} = \sum_{c=1}^{d} x_{ic} W^{q}_{c\pi(j)} = x_i \cdot \vec{w}^q_{\pi(j)} \tag{10}$$

$$K'_{ij} = \sum_{c=1}^{d} x_{ic} W^{k'}_{cj} = \sum_{c=1}^{d} x_{ic} W^{k}_{c\pi(j)} = x_i \cdot \vec{w}^k_{\pi(j)} \tag{11}$$

Now, consider the dot-product of rows from $Q$ and $K$:

$$(QK^T)_{ij} = \sum_{c=1}^{d_k} Q_{ic} K_{jc} \tag{12}$$

Similarly, for $Q'$ and $K'$:

$$(Q'K'^T)_{ij} = \sum_{c=1}^{d_k} Q'_{ic} K'_{jc} \tag{13}$$

Using the previous definitions of $Q'$ and $K'$, and plugging them into the above equation, we see:

$$(Q'K'^T)_{ij} = \sum_{p=1}^{d_k} \left( \sum_{c=1}^{d} x_{ic} W^{q'}_{cp} \right) \left( \sum_{r=1}^{d} x_{jr} W^{k'}_{rp} \right) \tag{14}$$

We take the Equation 10 11 into Equation 14 and we have:

$$\begin{aligned} (Q'K'^T)_{ij} &= \sum_{p=1}^{d_k} \left( x_i \cdot \vec{w}^q_{\pi(p)} \right) \left( x_j \cdot \vec{w}^k_{\pi(p)} \right) \\ &= \sum_{p=1}^{d_k} \left( x_i \cdot \vec{w}^q_p \right) \left( x_j \cdot \vec{w}^k_p \right) \end{aligned} \tag{15}$$

Because the permutation $\pi$ is applied consistently to both $W^q$ and $W^k$, and the dot product operation is commutative, each term of the form $x_i \cdot W^{q'}_{\pi(p)}$ will correspond to a term of the form $x_j \cdot W^{k'}_{\pi(p)}$, yielding identical products as in the original $QK^T$ computation.

Therefore, we can conclude that:

$$QK^T = Q'K'^T$$

Hence, the attention scores remain unchanged after the permutation.

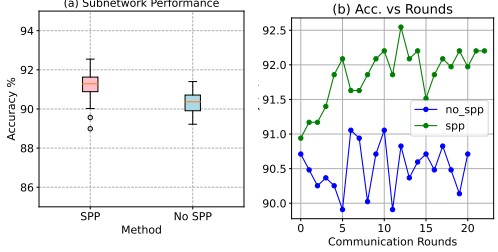

Figure 5: Local model performance salient parameter prioritization vs no salient parameter prioritization.

---

**Algorithm 1** Resource-aware Federated Learning

---

**Input:** Foundation model $\mathcal{F}(\mathcal{W}_g)$, clients set $S$, communication rounds $T$

**for** $t = 1, 2, \ldots, T$ **do**

    Salient Parameter Prioritization on $\mathcal{W}_g$

    Server establishes communication with participate clients $S^t \subseteq S$.

    $\tau^t \leftarrow$ clients resource constraints sets.

    $C^t \leftarrow$ sampled sub-network configuration satisfying $\tau^t$.

    **for** each client $s_\tau \in S^t$ and $c_\tau \in C^t$ **do**

        $\mathcal{F}(\mathcal{W}_{c_\tau}) \leftarrow \mathcal{F}(\mathcal{W}_g[: c_\tau])$

        Communicate submodel $\mathcal{F}(\mathcal{W}_{c_\tau})$ to client $s_\tau$.

    **end for**

    Clients perform local updates on private data.

    Clients communicate local optimized resource-aware submodels back to the server.

    Server aggregates local updates and computes the new global model:

$$\mathcal{W}_g = \sum_{c_\tau \in C_t} \left( \mathcal{W}_g[: c_\tau] + \eta_{c_\tau} \nabla \mathcal{W}_{c_\tau} \right)$$

**end for**

---

## A.2 ABLATION STUDY

RaFFM leverages the salient parameter prioritization (SPP) component to identify impactful weights in FMs. Hence, it's critical to evaluate the effect of salient parameter prioritization. The experiment is conducted on BERT-Large with SST-2 (Wang et al., 2018) Figure 5 shows the comparison of SPP and without SPP. Figure 5 (a) box plot shows the local sub-model performance range for randomly sampled 100 clients from supernet with the same resource constraints. It's evident that SPP produces subnetworks with higher performance. Figure 5 (b) shows the communication rounds vs global model (super-network) accuracy. RaFFM equipped with SPP, produced higher model performance and a more stable learning trend.

