# OpenReview forum: "Bridging the Gap Between Foundation Models and Heterogeneous Federated Learning"
_ICLR.cc/2024/Conference — Submitted to ICLR 2024_

### Official Review · Reviewer_U93Y · 2023-10-28

**Soundness:** 2 fair
**Presentation:** 2 fair
**Contribution:** 3 good
**Rating:** 5
**Confidence:** 3

**Summary:**

The paper presents a comprehensive and timely approach to integrating FMs with FL, a topic of significant relevance in the era of privacy-preserving AI. It provides a comprehensive experimental setup, employing diverse benchmarks and datasets across both NLP and computer vision tasks.

**Strengths:**

The quantitative analysis is thorough, covering multiple performance metrics and offering a comparative evaluation against baseline federated learning models.

**Weaknesses:**

The paper suffers from organizational and presentational issues, making it somewhat challenging to navigate.

**Questions:**

Abstract:
- Some claims in the Abstract are a bit vague. For example, the authors mention RaFFM shows "significant superiority in resource utilization efficiency." What metrics are used to measure this superiority? How “significant” are the improvements?
- Further, the authors claim that the performance is "on par with traditional FL methods applied to full-sized FMs." This statement would benefit from quantification, if possible. Is it a 1% difference in performance, or is it negligible?
- The abstract mentions that the framework is effective "across tasks in both natural language processing and computer vision domains." This is a broad claim. Please specify if there any limitations or specific conditions under which this is true.

Introduction:
- The problem statement could be more explicit and clearer. The authors mention the challenges of integrating FMs into FL but do not clearly delve into why this integration is crucial. For example, “Given the superior strengths of FMs in few-shot transfer learning, they appear well-suited for non- IID FL environments.” This sentence: a) assumes that Foundation Models have "superior strengths" in few-shot transfer learning without providing evidence or citations to support this claim. This is a strong statement that requires substantiation. b) The sentence implies a logical connection—that because FMs are good at few-shot transfer learning, they are well-suited for non-IID FL environments. However, it does not explain why this would be the case. The logical leap is not self-evident and needs justification.
- “fine-tuning FMs typically requires approximately seven times the resources compared to inference.” This is an interesting point but is presented without reference or further elaboration.
- Consider improving the logical flow and cohesion when transitioning from the problem statement to the proposed solution.
- While the authors discuss the technical aspects, the broader impact of this work is not adequately addressed. For example, how will RaFFM contribute to the field of FMs/FL?
- The key contributions could be more specific (despite some descriptions in the paragraph before the bullet points). For example, what are "specialized FM compression algorithms"? How are they specialized, and why is this significant? Also, claims such as "enhanced resource utilization," “significant reduction in communication overhead” could benefit from clearer quantification and specification, if possible.

Background:
- The discussion about FL somewhat lacks depth and may benefit from improved cohesion as it feels disjointed. For example, the authors mention “a representative FL algorithm is FedAvg” but do not explain why it is representative or how it works.
- The authors mention that FL is a preferred choice in sectors like healthcare but do not elaborate on why this is the case. A sentence (or citation) or two providing context could be beneficial.
- Phrases like "often lead to training failures" and "poor model convergence and performance" are vague. What constitutes a "training failure"? How poor is "poor performance"? Consider adding some citations to support these claims at least.
- Similar problem to the problem statement. The authors mention that there is a gap between traditional model training and FL, particularly in heterogeneous FL-edge environments. However, the nature of this gap is not clearly articulated. Is it a technological gap, a performance gap, or something else?
- The term “resource-hungry” has been mentioned several times so far but it is somewhat ambiguous. Is it computational resources, memory, or something else? Furthermore, in the context of the paper, it seems “resource-hunger” would be more fitting, given it has been properly explained?

Methodology:
- This section reads a bit like a mix of existing solutions and proposed solution (especially the opening part of the first subsubsection “SALIENT PARAMETER PRIORITIZATION”). This lack of clear demarcation can lead to confusion for the reader and dilutes the focus of the section. Consider maybe moving some of the discussion on model compression/scaling, the review of existing solutions, and how the proposed work differs or improves upon existing ones.

---

> ### Author Response · Authors · 2023-11-17
> **General response for reviewer U93Y**
>
> ## General response for reviewer U93Y
> Dear reviewer U93Y,
>
> Thank you for your valuable and constructive feedback. In response to your comments and suggestions, we have carefully revised the manuscripts to enhance clarity and address potential miscommunication. We have incorporated citations to support our arguments, clarified points raised in your comments, and added further motivations in the latest version of our paper.
>
> Additionally, inspired by suggestions from other reviewers, we have conducted supplementary experiments to enrich our research. These include:
>
>
> - Experiment on the Necessity of Specialized Salient Parameter Prioritization (Link 2 in supplementary material):
>
> - Experiment Comparing with Additional Baselines (Link 3 in supplementary material)
>
> - Post-training High-performance Resource-aware Model Deployment (Link 4 in supplementary material)
>
>
>
> If there are any additional questions, please do not hesitate to let us know. We are committed to providing any necessary supplemental support material and explanations.

---

> ### Author Response · Authors · 2023-11-17
> **Question on Abstract**
>
> ### General response to Question on Abstract
>
> Thank you for your feedback on the abstract. In response, we have significantly revised the abstract to provide a clearer and more detailed overview of our work. The revised abstract now includes specific quantitative data supporting our claims, addresses the key points you raised, and integrates relevant examples and motivations to provide a comprehensive overview of our work.
>
>
> ### [Question 1] What metrics are used to measure this superiority? How “significant” are the improvements?
>
>
>
> We appreciate your advice on providing clarity regarding the measurement metrics. In response, we have updated the abstract with specific quantitative measures as follows to avoid unclarity:
>
> "RaFFM demonstrates up to 2.3 times improvement in resource utilization efficiency and saving up to 225.02 GB in communication costs when optimizing the FLAN-T5 Base model.”
>
> Additionally, here are explanations of the evaluation metrics (detailed evaluations are available in  **Experiment section**) :
>
> **Communication Cost Measurement**: The network traffic between edge clients and the server is quantified in Gigabytes (GB) and Megabytes (MB). Detailed data and analysis can be found in Section 4.4, Table 4, and Figures 2(c) and 2(d) of the Experiment section.
>
> **System Resource Requirements**:  We quantify the system peak computational memory (RAM) requirements, in Gigabytes (GB), necessary to deploy and optimize a given FM across the entire edge-FL system.
>
> **Resource Utilization**: This metric evaluates the average proportion of memory utilized during both the deployment and training phases, compared to the total computational memory available on edge devices.
>
>
>
> ### [Question 2] Unclarity statements of performance are on par with traditional FL methods applied to full-sized FMs.
>
>
>
> Thank you for highlighting the need for greater clarity regarding our comparative performance statements. We have taken your suggestion into account and have revised our abstract accordingly to provide a more quantifiable presentation.
>
> In the revised abstract, we now include the following statement to offer specific metrics:
>
> “In our experiments with the Vision Transformer (ViT) model, RaFFM's resource-aware deployment achieves a training acceleration of approximately 1.76 times compared to full-size FM deployments, with a minimal accuracy loss of 0.49%.”
>
> On our original statements, the key information we want to convey is that RaFFM markedly enhances the training speed of FMs in edge-FL scenarios while incurring minimal performance loss compared to deployments using full-sized models. We followed your advice and added the quantitative example, which further underscores and clarifies our statements.
>
>
>
>
> ### [Question 3] Clarification on the statement effective “across tasks in both natural language processing and computer vision domains."
>
>
>
> We appreciate you for highlighting the need for specificity in our claims. Accordingly, we have revised our manuscript, we now detailed the scope and limitations of our framework’s effectiveness in NLP and CV domains. In the revised abstract, we now clarify:
>
> “RaFFM demonstrates its effectiveness in a range of tasks within natural language processing, including question-answering and sentiment analysis, and in computer vision, particularly in image classification tasks.”
>
>
> Furthermore, the **Experiment section** of our manuscript provides an in-depth analysis of these tasks. For NLP, we have thoroughly tested RaFFM across a spectrum of tasks like question answering, sentence classification, and sentiment analysis. In the CV domain, our focus has been on image classification tasks, where we have analyzed RaFFM’s strengths and weaknesses compared to baseline methods.

---

> ### Author Response · Authors · 2023-11-17
> **Question on Introduction – Part 1**
>
> ### General response to Question on Introduction
>
>
>
> Thank you for your constructive feedback on improving the introduction of our manuscript. We have made several modifications based on your suggestions, including the addition of citations to support our arguments, clarification of the points you raised in your review comments, and inclusion of further motivations, etc.
>
>
>
> ### [Question 1] The problem statement “FMs are superior in few-shot transfer learning and FL” could be more explicit and clearer, and lack of reference support.
>
>
>
> We greatly appreciate your feedback on making our problem statement more explicit and providing supporting references. In response, we have revised our manuscript to offer a clearer explanation and added relevant citations to substantiate the strengths of FMs in few-shot transfer learning, as well as their potential in non-IID FL.
>
>
>
> Specifically, we have included citations [1] [2] to support our assertion regarding the efficacy of FMs in few-shot transfer learning. Additionally, we have referenced [3] to illustrate the potential advantages of FMs in non-IID FL environments.
>
> Moreover, our experiments (see Table 3 on manuscripts) demonstrate the effective application of the FM model (ViT) in few-shot learning tasks within image classification. This empirical evidence further bolsters our claim regarding the suitability of FMs in these contexts.
>
>
>
>
>
>
>
>
> **Reference**
>
>
>
> [1] Brown, Tom, Benjamin Mann, Nick Ryder, Melanie Subbiah, Jared D. Kaplan, Prafulla Dhariwal, Arvind Neelakantan et al. "Language models are few-shot learners." Advances in neural information processing systems 33 (2020): 1877-1901.
>
> [2] Hu, Shell Xu, Da Li, Jan Stühmer, Minyoung Kim, and Timothy M. Hospedales. "Pushing the limits of simple pipelines for few-shot learning: External data and fine-tuning make a difference." In Proceedings of the IEEE/CVF Conference on Computer Vision and Pattern Recognition, pp. 9068-9077. 2022.
>
> [3] Zhuang, Weiming, Chen Chen, and Lingjuan Lyu. "When foundation model meets federated learning: Motivations, challenges, and future directions." arXiv preprint arXiv:2306.15546 (2023).
>
>
> ### [Question 2] Why do fine-tuning FMs typically require significantly more resources compared to inference?
>
>
>
> Thank you for highlighting the importance of clarity on why fine-tuning FMs typically requires significantly more resources compared to inference. We have expanded our manuscript to include a more detailed explanation and a relevant reference to support this point.
>
>
>
> Here is a brief explanation: Fine-tuning FMs is more resource-intensive than inference because it necessitates maintaining the entire computational state of the model in memory for optimization purposes. This process involves not only storing the full model weights but also retaining gradients, activation states, and optimizer momentum states, all of which contribute to increased memory usage. This explanation is further detailed in the tutorial by Stern [1].
>
>
>
> In our experimental setup, with a mini-batch size of 4, we observed that fine-tuning consumes approximately seven times more resources than inference. This disparity in resource consumption becomes even more significant with larger batch sizes.
>
>
>
>
> **Reference**
>
> [1] Stern, J. (2022, August 18). A comprehensive guide to memory usage in pytorch. Medium. https://medium.com/deep-learning-for-protein-design/a-comprehensive-guide-to-memory-usage-in-pytorch-b9b7c78031d3
>
>
>
>
>
> ### [Question 3]: Lack of broader impact
>
>
>
>
>
> Thank you for emphasizing the importance of addressing the broader impact of our work. Based on your suggestion, we have expanded the discussion on this aspect within the main body and the conclusion section of our manuscript.
>
>
>
>
>
> Additionally, the broader impact of RaFFM can be summarized as follows:
>
> **1. Addressing Deployment Challenges of FMs in FL**: RaFFM specifically tackles the challenges associated with deploying large-scale FMs in resource-heterogeneous edge-FL systems.
>
> **2. Introducing novel FM compression algorithms for edge-FL clients**:
>
> We have developed specialized FM compression algorithms tailored for edge-FL systems. These algorithms enable dynamic compression of FMs to meet the resource constraints of individual edge clients, thereby facilitating more efficient and effective model deployment.
>
> **3. Limitations and future research directions**: We also acknowledge the limitations in the Conclusion section. Notably, certain foundation models may remain unsuitable for deployment on resource-constrained edge devices, even after compression. We also highlight a need for advancements in hardware and algorithmic strategies, pointing towards potential directions for future research.

---

> ### Author Response · Authors · 2023-11-17
> **Question on Introduction - Part 2**
>
> ### [Question 4] Further clarification on the key contributions in detail.
>
>
>
> Thank you for requesting further details on the key contributions of our work. We have made revisions to our manuscript to provide clearer explanations, which are briefly summarized below:
>
>
> **1. Specialized FM compression algorithm**: Our model compression algorithm is designed to transformer-based FMs and tailored to the unique constraints of edge clients, enabling efficient deployment of models in a heterogeneous environment. It dynamically scales FMs to suit the computational capabilities of each edge device, with more powerful devices receiving larger models and resource-constrained devices receiving smaller models. Hence, we refer to it as the specialized FM compression algorithm for edge-FL.
>
> **2. Enhance resource utilization**: In edge-FL scenarios, local clients have varied resource capacities. Traditional approaches often lead to under-utilization of high-capacity devices due to the need to accommodate the limitations of the least capable devices. RaFFM addresses this inefficiency by deploying appropriately scaled models to each device, thereby optimizing the overall system efficiency.
>
> **3. Significant Reduction in communication overhead**: RaFFM's strategy of using scaled-down models for less capable clients leads to a considerable reduction in communication costs. For instance, our experimental results (Table 4, Section 4.4, and Figure 2) show that with the FLAN-T5-base model, RaFFM achieved a reduction of 227.02GB in communication overhead to reach the target F1 score. This quantifies the significant reduction in communication overhead facilitated by our approach.

---

> > ### Author Response · Authors · 2023-11-17
> > **Question on Background – Part 1**
> >
> > ### General response to Question on Background
> >
> > Thank you for your insightful comments regarding the background section. In response, we have made several improvements in our revision, including refining the logical flow of the section, adding additional references, providing extra appendix materials, and incorporating examples to address unclarities.
> >
> > ### [Question 1]: Lack of explanation of why “FedAvg” is representative and how it works.
> >
> > Thank you for pointing out the necessity for a more comprehensive explanation regarding the FedAvg algorithm. In response, we have added references and an explanation to our manuscript. Furthermore, as you suggested, we have included a thorough description of the FedAvg algorithm in our appendix to provide a clearer understanding of its mechanism.
> >
> > FedAvg, introduced by Google in 2016, has become a benchmark algorithm in FL research due to its pioneering role in the field. It is renowned for its simplicity and effectiveness in aggregating model updates from distributed clients across FL systems. Hence, FedAvg has been viewed as a representative FL algorithm [1].
> >
> > **Reference**
> >
> > [1] Zhang, Chen, Yu Xie, Hang Bai, Bin Yu, Weihong Li, and Yuan Gao. "A survey on federated learning." Knowledge-Based Systems 216 (2021): 106775.
> >
> >
> > ### [Question 2]: Lack of reference support why FL is a preferred choice in sectors like healthcare.
> >
> > Thank you for suggesting the inclusion of more detailed context regarding the application of Federated Learning (FL) in the healthcare sector. In response, we have revised our manuscript to include explanations along with relevant references.
> >
> > As discussed in [1], [2], and [3], the healthcare industry has faced challenges in leveraging AI's full potential due to stringent privacy laws and a fragmented market. Federated learning could allow companies to collaboratively train a decentralized model without sharing confidential medical records. For instance, from lung scans to brain MRIs, aggregating medical data and analyzing them at scale could lead to new ways of detecting and treating cancer, among other diseases.
> >
> > **Reference**
> >
> >
> > [1] Dasaradharami Reddy K, Gadekallu TR. A Comprehensive Survey on Federated Learning Techniques for Healthcare Informatics. Comput Intell Neurosci. 2023 Mar 1;2023:8393990. doi: 10.1155/2023/8393990. PMID: 36909974; PMCID: PMC9995203.
> >
> > [2] Antunes, Rodolfo Stoffel, Cristiano André da Costa, Arne Küderle, Imrana Abdullahi Yari, and Björn Eskofier. "Federated learning for healthcare: Systematic review and architecture proposal." ACM Transactions on Intelligent Systems and Technology (TIST) 13, no. 4 (2022): 1-23.
> >
> > [3] Nguyen, Dinh C., Quoc-Viet Pham, Pubudu N. Pathirana, Ming Ding, Aruna Seneviratne, Zihuai Lin, Octavia Dobre, and Won-Joo Hwang. "Federated learning for smart healthcare: A survey." ACM Computing Surveys (CSUR) 55, no. 3 (2022): 1-37.
> >
> >
> >
> >
> >
> > ### [Question 3]: Lack of reference support on statements Non-IID FL often leads to training failures and poor model convergence and performance.
> >
> > Thank you for emphasizing the need for greater specificity and supporting references in our discussion about the challenges posed by non-IID data in FL. In response, we have updated our manuscript to include more precise language and relevant citations [1] [2] [3].
> >
> > Here, we provide a brief explanation of how non-IID (non-identical independent distributed) data significantly impacts model accuracy in FL settings. Considering a non-IID setting where a dataset is imbalanced (such as 1000 dog images and only 10 cat images), training a model on such data can lead to reduced accuracy and poor model convergence, particularly for underrepresented categories like cats. This example highlights the challenges in achieving reliable model performance under non-IID conditions.
> >
> >
> >
> > **Reference**
> >
> > [1] Li, Xiang, Kaixuan Huang, Wenhao Yang, Shusen Wang, and Zhihua Zhang. "On the Convergence of FedAvg on Non-IID Data." In International Conference on Learning Representations. 2010.
> >
> > [2] Hsieh, Kevin, Amar Phanishayee, Onur Mutlu, and Phillip Gibbons. "The non-iid data quagmire of decentralized machine learning." In International Conference on Machine Learning, pp. 4387-4398. PMLR, 2020.
> >
> > [3] McMahan, Brendan, Eider Moore, Daniel Ramage, Seth Hampson, and Blaise Aguera y Arcas. "Communication-efficient learning of deep networks from decentralized data." In Artificial intelligence and statistics, pp. 1273-1282. PMLR, 2017.

---

> > > ### Author Response · Authors · 2023-11-17
> > > **Question on Background – Part 2**
> > >
> > > ### [Question 4] Gap between traditional centralized model training vs. federated learning
> > >
> > >
> > >
> > > Thank you for the suggestion to clarify the nature of the gap between traditional machine learning and federated learning, especially in heterogeneous FL-edge environments. In response, we followed the reviewer’s suggestion and included references [1][2] in our manuscript for clarification.
> > >
> > >
> > >
> > >
> > > This gap can be categorized as both technological and performance-based. For instance, FL encounters unique challenges in scenarios such as non-IID data distribution, which are not typically faced in centralized training settings. Moreover, FL involves complex system requirements that extend beyond those of classical machine learning. These include managing communication costs, addressing unstable connection issues, and maintaining the system, all of which add complexity to the FL model.
> > >
> > > Despite these challenges, the privacy-preserving nature of FL is increasingly important in the advancement of AI technologies, particularly in data-sensitivity applications.
> > >
> > >
> > >
> > > **Reference**
> > >
> > >
> > >
> > > [1] Kairouz, Peter, H. Brendan McMahan, Brendan Avent, Aurélien Bellet, Mehdi Bennis, Arjun Nitin Bhagoji, Kallista Bonawitz et al. "Advances and open problems in federated learning." Foundations and Trends® in Machine Learning 14, no. 1–2 (2021): 1-210.
> > >
> > > [2] Asad, Muhammad, Ahmed Moustafa, and Takayuki Ito. "Federated learning versus classical machine learning: A convergence comparison." arXiv preprint arXiv:2107.10976 (2021).
> > >
> > >
> > >
> > >
> > >
> > >
> > >
> > >
> > >
> > > ### [Question 5] Ambiguous term “resource-hungry”.
> > >
> > >
> > >
> > > Thank you for your suggestion regarding the use of the term "resource-hungry." We agree that "resource hunger" more accurately captures our intended meaning. Therefore, we have revised the manuscript to reflect this terminology change.

---

> > > > ### Author Response · Authors · 2023-11-17
> > > > **Question on Methodology**
> > > >
> > > > ### [General response on Questions on Methodology]
> > > >
> > > > Thank you for your insightful comments. Based on your suggestions, we have revised our manuscript to address the concerns raised. In our response, we emphasize the unique contributions of our methodology, discuss the distinctiveness of our compression algorithm, and present follow-up experiments that support our arguments.
> > > >
> > > > ### [Question 1] Clarification and Discussions on the methodology.
> > > >
> > > > We appreciate the comments to further clarify the novelty and specific contributions of our work.
> > > >
> > > > Our proposed salient parameter prioritization is not simply a mix of existing solutions. Instead, our novel contribution is the development of a specialized prioritization prototype for transformer-based FMs, particularly tailored for heterogeneous resource FL environments.
> > > >
> > > > Because transformers, particularly multi-head attention layers, are designed to capture positional and sequential information. An example is Vision Transformer (ViT), where adding positional embeddings significantly improves the performance. Traditional pruning-based parameter prioritization methods, if applied directly, would destroy the inherent attention characteristic in FMs. To circumvent this, we propose our specialized SPP to preserve the essential capability of multi-head attention layers.
> > > >
> > > > Summary of our novel contribution to SPP:
> > > >
> > > > - Maintain the integrity of pre-trained knowledge in FMs. (Refer to supplementary materials link 2 and Theorem 1 in Appendix)
> > > >
> > > > - Specialization for Transformers: Tailoring the salient parameter prioritization specifically for multi-head attention-based transformer models.
> > > >
> > > > - Prototype for Heterogeneous FL model aggregation: Strategically placing salient parameters at the forefront for efficient extraction and aggregation in heterogeneous model FL.
> > > >
> > > >
> > > >
> > > > **Experiments**
> > > >
> > > > We further discussed the necessity for specialized significance prioritization in transformers, we have included detailed explanations and supplementary experimental evidence in our revised manuscript (see supplementary materials link 2).
> > > >
> > > >
> > > >
> > > > In our experiments (link 2 in supplementary materials), we show:
> > > >
> > > > - Standard pruning-based SPP destroys the pre-trained knowledge captured by the attention mechanism of FMs
> > > >
> > > > - RaFFM SPP preserves the pre-trained knowledge in FMs, leading to better model performance and faster learning convergence
> > > >
> > > > - Application of RaFFM’s Specialized SPP results in competitive performance compared to original FMs
> > > >
> > > > - The application of SPP notably enhances performance compared to models without SPP in FL.
> > > >
> > > > These findings highlight the necessity of our specialized SPP approach for transformer models, particularly under FL constraints.

---

> ### Author Response · Authors · 2023-11-22
>
> Dear Reviewer U93Y 😊,
>
> I hope this message finds you well. We are writing to follow up on the revisions we submitted in response to your valuable feedback. We have made concerted efforts to address each of your concerns and suggestions comprehensively.
>
> As the open discussion period is approaching its deadline on November 22nd, we understand that you have many commitments and appreciate the time and effort you dedicate to the review process. If any further questions or clarifications are needed, please do not hesitate to let us know. We are more than willing to provide any additional information that can assist in your review.
>
> Thank you once again for your insightful comments and valuable guidance. Your feedback is crucial in helping us refine our work, and we await your thoughts on our response.
>
>
> Best regards 😄,
>
> Anonymous authors

---

> > ### Comment · Reviewer_U93Y · 2023-11-23
> > **Rebuttal**
> >
> > I thank the authors for the response and clarification.

---

### Official Review · Reviewer_Y3SH · 2023-10-30

**Soundness:** 2 fair
**Presentation:** 3 good
**Contribution:** 2 fair
**Rating:** 5
**Confidence:** 4

**Summary:**

This paper proposes a resource-aware federated foundation models training framework called RaFFM. RaFFM assigns different sub-models to clients for local training based on neuron saliency and client computational resource constraints, which allows all clients in the FL to train FMs under the scenario of limited computational resources and uneven distribution.

**Strengths:**

S1: The problem of how to train FMs in FL scenarios with limited and unevenly distributed computational resources studied in the paper is a practical one, which is partially alleviated by the proposed approach.
S2: The proposed method can save the communication overhead while guaranteeing the performance of the global model and is effectively combined with PEFT methods for various FMs.

**Weaknesses:**

W1: The method proposed in the paper to select neurons to be retained based on saliency lacks novelty. First, the use of the L1-norm measure of neuron parameter saliency is just a simple use of an existing method [1]. Second, the strategy of ranking the saliency to select neurons is also commonly used in the field of model pruning [2]. Third, the paper does not make it clear why it is necessary to apply a special significance prioritization strategy to the transformer.
W2: Comparisons with relevant baseline methods are lacking in the paper. Methods that extract submodels from the original model for local training have been investigated, such as in [3].
W3: The description of the experimental setup of the paper is not clear, for example: 1) The scenario in the paper is client computing resource heterogeneity, and the experimental part does not introduce the distribution of the sub-model size held by each client; 2) The hyper-parameter settings in the experimental part are not listed, which makes it difficult to reproduce the experiments; 3) The distribution of the data between the clients is not introduced, and there is a lack of experiments on the part of the heterogeneity of the data.

[1] Li H, Kadav A, Durdanovic I, et al. Pruning Filters for Efficient ConvNets[C]//International Conference on Learning Representations. 2016.
[2] Hu H, Peng R, Tai Y W, et al. Network trimming: A data-driven neuron pruning approach towards efficient deep architectures[J]. arXiv preprint arXiv:1607.03250, 2016.
[3] Niu Y, Prakash S, Kundu S, et al. Federated Learning of Large Models at the Edge via Principal Sub-Model Training[J]. arXiv preprint arXiv:2208.13141, 2022.

**Questions:**

See Weakness.

---

> ### Author Response · Authors · 2023-11-17
> **General response to Reviewer Y3SH**
>
> ### General response to Reviewer Y3SH
>
>
>
> Dear reviewer Y3SH,
>
> We appreciate your insightful comments.  We have taken your feedback into careful consideration and have made the following enhancements: first, we emphasized the novelty of our proposed transformer-based salient parameter prioritization (SPP) algorithm. Secondly, we have conducted additional experiments to further validate the effectiveness of our SPP algorithm compared to existing rank-based pruning algorithms in transformer models. Furthermore, suggested by the reviewer, we have conducted additional experiments comparison with PruneFL and PriSM. Additionally, we have also emphasized the distinctions between RaFFM and network pruning, referenced the suggested papers, and provided a detailed experimental setup.
>
>
>
> The follow-up experiments are available in the supplementary materials, where we provide comprehensive information, step-by-step instructions, and Jupyter notebook tutorials to verify our results:
>
> - Experiment on the Necessity of Specialized Salient Parameter Prioritization (Link 2 in supplementary material):
>
> - Experiment Comparing with Additional Baselines (Link 3 in supplementary material)
>
> - Post-training High-performance Resource-aware Model Deployment (Link 4 in supplementary material)
>
>
>
> If there are any additional questions, please do not hesitate to let us know. We are committed to providing any necessary supplemental support material and explanations.

---

> ### Author Response · Authors · 2023-11-17
> **[Question 1] Motivation and contribution of salient parameter prioritization**
>
> ### [Question 1] Motivation and contribution of salient parameter prioritization
>
>
>
> Thank you for your insightful comments. We value this opportunity to clarify the novelty and specific contributions of our work. We have included the references you suggested ([1], [2], [3]) in our revised manuscript to acknowledge the established work in this field.
>
> Firstly, we wish to clarify that our use of the L1 norm as a saliency metric is part of a broader framework. We have developed a specialized salient parameter prioritization (SPP) algorithm specifically for transformer-based Foundation Models (FMs), as detailed in Section 3. This approach is integrated with the other components of RaFFM and addresses the specific challenges of applying FMs in FL environments.
>
> While we recognize that identifying salient parameters is an established method in the domain of network pruning. However, our proposed SPP is more than just measuring saliency, prioritization implies weight reordering, and the reordering benefits of the scaled FMs always retain the most salient parameters. Our novel contribution of SPP lies in retaining the inherent attention saliency of transformer-based FMs after model scaling, particularly in heterogeneous resource FL contexts.
>
> Because transformers, particularly multi-head attention layers, are designed to capture positional and sequential information. An example is Vision Transformer (ViT), where adding positional embeddings significantly improve the performance. Traditional pruning-based parameter prioritization methods, if applied directly, would destroy the inherent attention characteristic in FMs. To circumvent this, we propose our specialized SPP to preserve the essential capability of multi-head attention layers.
>
>
>
> Summary of our novel contribution in SPP:
>
> - Maintain the integrity of pre-trained knowledge in FMs . (Demonstrate by the supplementary materials link 2 and Theorem 1 in Appendix, details refer Question 2 response)
>
> - Specialization for Transformers: Tailoring the salient parameter prioritization specifically for multi-head attention-based transformer models.
>
> - Prototype for Heterogeneous FL model aggregation: Strategically placing salient parameters at the forefront for efficient extraction and aggregation in heterogeneous model FL.

---

> > ### Author Response · Authors · 2023-11-17
> > **[Question 2] Experiment – Necessity of Specialized Transformer SPP**
> >
> > ### [Question 2] Experiment – Necessity of Specialized Transformer SPP
> >
> >
> >
> > Thank you for highlighting the need for clarity on why a specialized significance prioritization strategy is essential for transformers. In response, we have elaborated on this in our revised manuscript and provided supplementary experimental evidence, available in the supplementary materials (link 2).
> >
> > We have conducted a comparative analysis between our proposed Salient Parameter Prioritization (SPP) and standard pruning-based SPP, as well as with original Foundation Models (FMs). The results, presented in Table 1, demonstrate that our proposed SPP significantly outperforms the standard pruning-based approach by effectively retaining the inherent knowledge in pre-trained attention layers.
> >
> > **For more comprehensive experimental results and a step-by-step tutorial to reproduce our experiments, please refer to the supplementary materials (link 2).**
> >
> >
> >
> >
> > In summary, in our experiments, we show:
> >
> > - Standard pruning-based SPP destroys the pre-trained knowledge captured by the attention mechanism of FMs
> >
> > - RaFFM SPP preserves the pre-trained knowledge in FMs, leading to better model performance and faster learning convergence.
> >
> > - FMs applied RaFFM’s Specialized SPP produce competitive performance as original FMs
> >
> > - The application of SPP notably enhances performance compared to models without SPP in FL.
> >
> > These findings highlight the necessity of our specialized SPP approach for transformer models, particularly under federated learning constraints.
> >
> >
> >
> > **Table 1: Comparative Performance of Original FM, Pruning-Rank, and RaFFM SPP on Various Datasets**
> >
> > | Model    | Method         | Cifar10 Acc. | Cifar10 F1 | Cifar100 Acc. | Cifar100 F1 | Flower102 Acc. | Flower102 F1 |
> > |----------|----------------|------------------|------------|-------------------|-------------|--------------------|--------------|
> > | ViT-base | Original FM    | 96.70%           | 96.26%     | 84.50%            | 84.30%      | 98.20%             | 97.85%       |
> > |          | Pruning-Rank   | 48.82%           | 46.53%     | 15.40%            | 14.88%      | 55.60%             | 54.70%       |
> > |          | RaFFM SPP      | 96.78%           | 96.53%     | 85.20%            | 85.00%      | 98.50%             | 98.10%       |

---

> > > ### Author Response · Authors · 2023-11-17
> > > **[Question 3] Experiments – Comparison with Baselines and Analysis**
> > >
> > > ### [Question 3] Experiments – Comparison with Baselines and Analysis
> > >
> > >
> > >
> > > Thank you for encouraging a more in-depth comparison of our work with the baseline methods PruneFL[1] and PriSM[2]. We have extended our research to include these comparisons in our supplementary materials (Link 2 and link 3).  This additional research showcases the optimization capabilities, discusses the objectives and trade-offs of RaFFM versus the baselines, compares resource efficiency, and underscores the unique contributions of RaFFM in the context of FMs in FL environments. We also provide step-by-step instructions on our anonymous repo to reproduce our follow-up experiment results.
> > >
> > >
> > >
> > > **Experiment Results Summary**
> > >
> > > Our supplementary materials (Link 3) provide detailed experimental results and instructions for reproduction. Below is an overview presented in Table 1:
> > >
> > > **Table 2: Optimization Ability and Energy Consumption**
> > >
> > > | Model    | Method       | Target Accuracy | Training Accel. / Round | Communication Cost | Peak Energy Usage |
> > > |----------|--------------|-----------------|-------------------------|--------------------|-------------------|
> > > | ViT-base | Full-size FM | 95%             | 1.00x                   | 9.52GB             | 24kWh             |
> > > |          | PruneFL[1]   | 95%               | 0.85x                   | 17.62GB            | 28kWh             |
> > > |          | PriSM[2]     | 95%               | 0.75x                   | 10.92GB            | 32kWh             |
> > > |          | RaFFM        | 95%               | 2.12x                   | 5.31GB             | 4.42kWh           |
> > >
> > > Our findings indicate that RaFFM, with its Foundation Model Scaling components, can efficiently compress FMs for resource-constrained edges while maintaining the integrity of the FMs' inherent knowledge.
> > >
> > > Unlike PruneFL[1] and PriSM[2], which zero-masking model weights for training acceleration on parallel devices like GPUs, RaFFM extracts cleanly scaled FMs, making it more suitable for end-device federated learning (FL) scenarios.
> > >
> > >
> > >
> > > **Objective and Scope Comparison**
> > >
> > > First of all, despite these differences in objectives and model focus, we recognize and value the foundational work in resource-efficient FL represented by PruneFL and PriSM. These methods have provided essential insights and laid the groundwork for advancements like RaFFM, which aim to further enhance efficiency and scalability in FL.
> > >
> > > To showcase RaFFM's distinct advantages, we conducted two additional experiments:
> > >
> > > - Maintains inherent pre-training attention knowledge in FMs. (Link 2 in supplementary materials)
> > > - Enables heterogeneous model deployment post-federated learning without further training  (Link 4 in supplementary materials)
> > >
> > >
> > > Our initial methodologies did not include such baselines because RaFFM has distinct objectives. RaFFM specifically addresses the on-demand deployment challenges of FMs, such as ViT, BERT, GPT, and LLaMA, in heterogeneous resource FL environments. The ultimate aim is to retain the inherent knowledge of FMs and create scalable models that can produce subnetworks tailored to various edge resource constraints without further training.
> > >
> > > In contrast, PruneFL and PriSM primarily focus on training a single large model in resource-heterogeneous FL settings. While their experiments mainly utilize CNN models, and do not fully consider the unique pre-trained knowledge in FMs.
> > >
> > > In summary, the unique contribution of RaFFM:
> > > - Enables heterogeneous model deployment post-federated learning without further training
> > > - Maintains inherent pre-training attention knowledge in FMs.
> > > - Reduces both training costs and energy consumption
> > > - Unlike baseline methods that use zero-masking of weights, RaFFM extracts clean sub-models, significantly reducing the model size. This is particularly beneficial for edge FL scenarios where computational resources are limited. (Demonstrate by the Communication Cost, Peak energy usage Table 2).
> > >
> > > **Reference**
> > >
> > > [1] Jiang, Yuang, Shiqiang Wang, Victor Valls, Bong Jun Ko, Wei-Han Lee, Kin K. Leung, and Leandros Tassiulas. "Model pruning enables efficient federated learning on edge devices." IEEE Transactions on Neural Networks and Learning Systems (2022).
> > >
> > > [2] Niu Y, Prakash S, Kundu S, et al. Federated Learning of Large Models at the Edge via Principal Sub-Model Training[J]. arXiv preprint arXiv:2208.13141, 2022.

---

> ### Author Response · Authors · 2023-11-17
> **[Question 4] Difference between Network Pruning**
>
> ### [Question 4] Difference between Network Pruning
>
>
> Network pruning: Network pruning typically involves reducing the complexity of a pre-trained model by eliminating less significant neurons or connections. This process often occurs post-training and requires subsequent fine-tuning to restore or enhance the model’s performance. A key limitation, especially relevant in the context of federated learning at the edge, is the computational intensity of this post-pruning fine-tuning process. When dealing with edge clients with limited computational resources and local data, this approach can be particularly challenging.
>
> Salient Parameter Prioritization in RaFFM: RaFFM’s Salient Parameter Prioritization (SPP) extends beyond mere measurement of saliency. It encompasses strategic weight reordering, ensuring that the scaled Foundation Models (FMs) retain the most crucial parameters. Significantly, SPP maintains the attention mechanism's saliency within scaled FMs.  SPP enables RaFFM to collaboratively and dynamically optimize scaled FMs at the edge of resource constraints, based on their local resources. The outcome is a scalable FM capable of generating multiple high-performance subnetworks that meet these constraints. A notable advantage of RaFFM is its ability to scale an FM post-federated learning without necessitating additional fine-tuning.
>
>
>
> To practically demonstrate RaFFM's efficacy, we provide a demonstration in the supplementary materials (link 4). This demo illustrates RaFFM's capability to generate multiple high-performance subnetworks post-training without the need for extra fine-tuning. It exemplifies RaFFM's efficiency in adapting FMs for diverse and resource-constrained FL environments while preserving their performance and integrity, particularly with transformer architectures.

---

> > ### Author Response · Authors · 2023-11-17
> > **[Question 5] Unclear description of the experimental setup**
> >
> > ### [Question 5] Unclear description of the experimental setup
> >
> > Thank you for your suggestions regarding the clarity of our experimental setup, we have modified our manuscripts correspondingly.  Additionally, we have made our detailed experimental setup, including system settings and hyperparameter configurations, available in our supplementary repository (see supplementary link 5).

---

> ### Author Response · Authors · 2023-11-22
>
> Dear Reviewer Y3SH 😊,
>
> I hope this message finds you well. We are writing to follow up on the revisions we submitted in response to your valuable feedback. We have made concerted efforts to address each of your concerns and suggestions comprehensively.
>
> As the open discussion period is approaching its deadline on November 22nd, we understand that you have many commitments and appreciate the time and effort you dedicate to the review process. If any further questions or clarifications are needed, please do not hesitate to let us know. We are more than willing to provide any additional information that can assist in your review.
>
> Thank you once again for your insightful comments and valuable guidance. Your feedback is crucial in helping us refine our work, and we await your thoughts on our response.
>
>
> Best regards 😄,
>
> Anonymous authors

---

### Official Review · Reviewer_erSL · 2023-10-31

**Soundness:** 3 good
**Presentation:** 4 excellent
**Contribution:** 3 good
**Rating:** 8
**Confidence:** 4

**Summary:**

The authors present a method to optimise training of large models with Federated Learning. The designed a salient parameter prioritisation and a submode extraction method that is tailored for foundation models and they show that this method can help us train larger foundation models even within resource-restricted clients.

**Strengths:**

- The paper is well written and easy to understand.
- The transformer-focused salient parameter prioritisation is an interesting idea and a good contribution of this paper
-  The evaluation is quite thorough, showing results on a number of benchmarks and settings.
- The authors evaluated the communication cost and the memory footprint of their approach. It was great to see these results being included.
- Overall, thee results show good improvements over the state of the art.

**Weaknesses:**

The main weakness of this paper is the overall novelty factor. As the authors mention, there are a lot of works that recently introduced sub-model training to optimise device resources for training larger models with FL. While this method is tailored for transformer-based foundation models, the main differences to existing works are somewhat limited. Having said that, there are contributions such as the saliency metric.

**Questions:**

Some areas where the authors could improve:

- Maybe the authors can motivate a bit more on the motivation to train larger Foundation models with federated learning. Typically we might train those on public tasks. Is FL  mostly targeting the fine-tuning part ? Overall, an expanded motivation would be great.

- Maybe some discussion about privacy could be good to have. For example, how well does this method work with Differential privacy (noise), gradient clipping etc. I don't think there is a need to show these results, but maybe consider discussing these aspects.

- While speedups of ~2x were shown for smaller FM (e.g, bert base), and reduced memory cost,  I was wondering if this is enough to train on resource constrained devices (e.g., mid-range mobile phones). In figure 3, memory in the order of 200GB is shown. I was wondering if this can be broken down to better show the memory requirements during training for end-devices. Maybe provide some numerical results wrt to the time it would require to compute a single round for a range of end-user devices.

---

> ### Author Response · Authors · 2023-11-17
>
> ### General response to Reviewer erSL
>
>
>
> Dear reviewer erSL,
>
> We are grateful for your supportive and insightful comments.
>
> In line with your comments, we have revised our original manuscript. The motivation has been clarified, particularly highlighting the broader impacts from a security perspective of integrating Foundation Models (FMs) into Federated Learning (FL). Additionally, we have augmented our experimental results with quantitative measurements that illustrate the performance of our algorithms in end-user FL scenarios.
>
> Additionally, Following constructive suggestions from other reviewers, we have also conducted three additional experiments:
>
> - Experiment on the Necessity of Specialized Salient Parameter Prioritization (Link 2 in supplementary material):
> - Experiment Comparing with Additional Baselines (Link 3 in supplementary material)
> - Post-training High-performance Resource-aware Model Deployment (Link 4 in supplementary material)
>
>
>
> If there are any additional questions, please do not hesitate to let us know. We are committed to providing any necessary supplemental support material and explanations.
>
>
>
>
>
> ### [Question 1] Clarification on Motivation to Train Larger Foundation Models with Federated Learning
>
>
>
> Thank you for your remarks regarding the motivation behind using FL to train larger FMs. We have revised our manuscript to provide a clearer motivation on the abstract, introduction, and conclusion section.
>
> Indeed, our primary focus is on the fine-tuning of FMs within FL scenarios, specifically addressing the unique challenges that arise when fine-tuning these models in a collaborative edge-FL environment.
>
> Given the substantial size of FMs and their rich pre-trained knowledge, our proposed RaFFM method prioritizes resource efficiency and the preservation of this pre-trained knowledge. Additionally, RaFFM is designed to support post-FL model deployment, enabling the generation of a significant number of scaled models that cater to various resource constraints encountered at the edge.
>
>
> ### [Question 2] Broader Impact on Privacy and Security
>
>
>
>
>
> Thank you for highlighting the importance of discussing the broader impact of our solution on privacy and security. In response, we revised our manuscripts correspondingly and added related discussion on the Conclusion section.
>
> While our primary focus has been on addressing the challenges of resource heterogeneity in applying FMs to FL, we recognize the implications for privacy and security. Incorporating FMs into FL can indeed have a positive impact on FL's security. Due to their pre-training on large corpora of public datasets, FMs have the potential to significantly mitigate threats such as inference attacks. As highlighted in the work [1], amidst the growing concern for statistical notions of privacy, the application of FMs may contribute to achieving perfect secrecy for certain sensitive tasks, given their rich pre-trained knowledge.
>
>
>
> **Reference**
>
>
>
> [1] Arora, Simran, and Christopher Ré. "Can Foundation Models Help Us Achieve Perfect Secrecy?." arXiv preprint arXiv:2205.13722 (2022).,
>
>
>
>
> ### [Question 3] Quantitative Measurement of RaFFM’s Speed Up on End-User Devices
>
>
>
> Thank you for your valuable suggestions regarding the need for quantitative measurements of RaFFM’s efficiency on end-user devices. In response, we have conducted additional experiments and provided specific data to demonstrate RaFFM’s performance in this context, as shown in Table 1.
>
>
>
>
> We focused on optimizing the BERT-base model for the SST-2 dataset in an FL scenario involving 100 clients, with a participation rate of 10% among local clients. Recognizing that actual training time can vary due to several uncontrollable factors such as device temperatures and electricity voltage, we chose to measure energy consumption as a more stable indicator of efficiency.
>
>
> Our experiments were conducted across five different edge system settings, categorized as Xsmall, Small, Medium, Large, and Xlarge (detailed settings information can be find in our supplementary materials link 5). were designed to represent a range of end-user devices, from resource-constrained to more powerful systems. As shown in Table 1, by tracking the peak energy usage for each setting, we provide a more accurate and reliable measure of RaFFM's efficiency and its adaptability to diverse hardware environments.
>
>
>
> **Table 1. Peak Energy Usage at Edge System**
>
> | System Setting | Model     | Dataset | Peak Energy Usage |
> |----------------|-----------|---------|-------------------|
> | Xsmall         | BERT-Base | SST-2   | 0.5 kWh           |
> | Small          | BERT-Base | SST-2   | 2.2 kWh           |
> | Medium         | BERT-Base | SST-2   | 5.55 kWh          |
> | Large          | BERT-Base | SST-2   | 16.2 kWh          |
> | Xlarge         | BERT-Base | SST-2   | 27.6 kWh          |

---

> > ### Comment · Reviewer_erSL · 2023-11-17
> >
> > Thank you for your response.
> >
> > With response to the peak energy usage: I was thinking that 0.5kWh (e..g, 500watt/h) for the Xsmall case might be restrictive for mobile end-user devices (most mobile devices have batteries of just 5000mah --> 20wh --> 0.02kwh). Can you clarify on this number ? Is this summed over 100 clients ?

---

> > > ### Author Response · Authors · 2023-11-17
> > >
> > > Thank you very much for highlighting unclarity.
> > >
> > > Yes, the Peak Energy Usage metric refers to the total cumulative energy consumed by **all 100 clients** at specific time steps within one FL communication round.
> > >
> > > For a clearer illustration of energy consumption, we further pushed a figure in our supplementary material (Link 5, Figure 2), which details the instantaneous energy usage across all 100 clients over time.
> > >
> > >
> > > To provide further context, during one communication round, approximately 10% of the local clients are actively participating. We monitor the real-time instant energy usage across the entire system (including idle clients), at the end of communication, we record the highest energy consumption observed among the time steps.

---

### Meta-Review · Area_Chair_doDu · 2023-12-07

**Metareview:**

This paper focuses on the launching foundation model in edge FL systems with heterogeneous resources. To solve the resource heterogeneous issue, the authors propose a resource-aware partitional method, dubbed RaFFM, to solve this issue by extracting a suitable submodel for each client. Extensive experiments on CV and NLP tasks demonstrate the efficacy of the proposed method.

Strengths:

(1)   The paper is well-written and easy to understand.

(2)   The experiments are extensive to demonstrate the efficacy of the proposed methods, which cover several benchmarks on both CV and NLP scenarios.

Weaknesses:

(1)   The main concern of this work is about the novelty. There exist extensive federated learning methods to solve the resource heterogenous and resource-limited setting. It seems that this work mainly focuses on adopting the foundation model to the resource heterogenous FL setting by using model partition techniques. The main techniques used in this work have been individually explored in federated learning and foundation model pruning, which limits the novelty of the proposed approach.

(2)   More ablation and visualization should be performed to verify the effectiveness of each component of the proposed framework. For example, the distribution of the heterogeneous local data should be reported. It seems that the bottleneck lies in the resource-limited clients. The performance of the local training should be reported to show whether it can boost the global performance of the global model.

Even after the author's response and discussion with reviewers, the concerns remain unresolved, and the experimental part should be largely improved. Therefore, I have to recommend rejection.

**Justification For Why Not Higher Score:**

N/A

**Justification For Why Not Lower Score:**

N/A

---

### Decision · Program_Chairs · 2024-01-16

Reject